# Discordant congenital heart defects in monochorionic twins: Risk factors and proposed pathophysiology

**Helia Imany-Shakibai[1], Ophelia Yin[2], Matthew R. Russell[3], Mark Sklansky[1,4], Gary Satou[1,4], Yalda Afshar●[1,2]***

**1** David Geffen School of Medicine, UCLA, Los Angeles, California, United States of America, **2** Division of Maternal Fetal Medicine, Department of Obstetrics and Gynecology, UCLA, Los Angeles, California, United States of America, **3** Department of Pediatrics, Kaiser Permanente Southern California, Los Angeles, California, United States of America, **4** Division of Pediatric Cardiology, UCLA Mattel Children's Hospital, Los Angeles, California, United States of America

* YAfshar@mednet.ucla.edu

**Data Availability Statement:** All individual level relevant de-identified data are within the paper (Tables).

## Abstract

A six-fold increase in congenital heart defects (CHD) exists among monochorionic (MC) twins compared to singleton or dichorionic twin pregnancies. Though MC twins share an identical genotype, discordant phenotypes related to CHD and other malformations have been described, with reported rates of concordance for various congenital anomalies at less than 20%. Our objective was to characterize the frequency and spectrum of CHD in a contemporary cohort of MC twins, coupled with genetic and clinical variables to provide insight into risk factors and pathophysiology of discordant CHD in MC twins. Retrospective analysis of all twins receiving prenatal fetal echocardiography at a single institution from January 2010 –March 2020 (N = 163) yielded 23 MC twin pairs (46 neonates) with CHD (n = 5 concordant CHD, n = 18 discordant CHD). The most common lesions were septal defects (60% and 45.5% in concordant and discordant cohorts, respectively) and right heart lesions (40% and 18.2% in concordant and discordant cohorts, respectively). Diagnostic genetic testing was abnormal for 20% of the concordant and 5.6% of the discordant pairs, with no difference in rate of abnormal genetic results between the groups (p = 0.395). No significant association was found between clinical risk factors and development of discordant CHD (p>0.05). This data demonstrates the possibility of environmental and epigenetic influences versus genotypic factors in the development of discordant CHD in monochorionic twins.

## Introduction

Congenital heart defects (CHD) are the most prevalent group of congenital anomalies, affecting approximately 0.9% of all singleton births [1, 2]. In monochorionic (MC) twins, the prevalence of CHD is six times higher, affecting 59 per 1000 live births [1].

Despite sharing an identical genotype, MC twins can develop discordant phenotypes for congenital malformations including CHD. Studies have not found strong genetic influences

**Funding:** Y.A. is funded by the Reproductive Scientists Development Program of the National Institute of Child Health and Human Development, National Institute of Health K12 HD000849.

**Competing interests:** The authors have declared that no competing interests exist.

on discordant CHD in MC twins, with attributable genetic causes epigenetic in origin as opposed to differences in germline mutations or different phenotypic expression of the same genotype [3, 4]. It has been hypothesized that environmental influences such as teratogens can interfere with epigenetic processes leading to differential gene expression and discordant CHD [5].

In addition to epigenetics, local placental influences have also been identified as significant factors contributing to discordant CHD [5]. A 2011 study reported that 41% of all studied cases of discordant CHD resulted from placenta-related pathophysiologic mechanisms [5]. It has been hypothesized that placental inter-twin vascular connections, cord insertion sites, and relative placental-share contribute to imbalance of blood flow leading to relative hypoperfusion of one twin [5].

A major risk factor unique to MC twin population is the development of twin-to-twin transfusion syndrome (TTTS), an anomaly of placental vascular anastomoses causing an imbalance of blood flow from the donor twin to the recipient twin. The rate of CHD rises to 9.3% in MC pregnancies complicated by TTTS [6]. In the setting of TTTS, the imbalance of blood flow and increased aortic velocity in the recipient twin leads to increased risk of structural CHD in the recipient twin compared to the donor twin [5, 7]. Although several factors have been linked to the development of discordant CHD in MC twins, there is currently an incomplete understanding of the pathophysiology and associated risk factors of this diagnosis [3, 5].

The importance of early identification for CHD has been demonstrated by previous studies [8–11]. Although many cases critical CHD diagnoses occur either prenatally or subsequently through newborn screening, one in five cases of critical CHD are not diagnosed until after the fourth week of life [10]. Despite advances in fetal echocardiography, up to 60% of CHD diagnoses are diagnosed postnatally [12, 13]. Late diagnosis of critical CHD is associated with increased risk of morbidity and mortality, increased hospital length of stay, and 35% higher inpatient costs during infancy [8, 9, 11]. Therefore, understanding the pathophysiology and risk factors associated with discordant CHD in MC twins is critical to allow for early evaluation of an at-risk fetus. Early detection would work to reduce delivery and postnatal complications and aid in prevention through identification of modifiable risk factors.

This study examines fetal and maternal variables comparing MC twin pairs discordant versus concordant for CHD. We aim to describe the spectrum of lesions in this population and to elucidate risk factors and pathophysiology of discordant CHD in a modern cohort of MC twins. The monochorionic twin pair serves as an ideal model to evaluate environmentally triggered errors in cardiac development that contribute to CHD, both in the multifetal pregnancy as well as singleton pregnancies.

## Methods

This retrospective cohort study utilized all prenatal screening echocardiograms conducted at the University of California, Los Angeles (UCLA) from January 2010 –March 2020 to identify all MC twins with concordant and discordant CHD with outcome of a livebirth. Dichorionic and conjoined twins were excluded from this study. Institutional Review Board (IRB) was obtained from UCLA (IRB #17–000925). Patient medical record numbers were used to obtain demographic information and were linked to the neonatal medical record numbers. Additional variables related to known risk factors for CHD independent of chorionicity, such as advanced maternal age, family history of CHD, high maternal pre-pregnancy body mass index (BMI), diabetes, and conception with assisted reproductive technology (ART) were also abstracted [14, 15]. All chart-abstracted information was stored in a de-identified research

database. Chorionicity was confirmed by both an early perinatal ultrasound and by review of final placental pathology after delivery. CHD diagnoses were determined by postnatal echocardiography along with surgical operative reports, if available.

Concordant CHD was defined as a twin pair with an identical CHD diagnosis. Discordant CHD was defined as twin pairs with one affected and one unaffected fetus or twin pairs with different CHD diagnoses. Chart review was conducted to collect genetic test data, CHD diagnoses, extra-cardiac anomalies, and invasive (clinical) genetic testing results were obtained from prenatal amniocentesis samples or postnatal serum samples. Specifically, genetic test modalities included in analysis were karyotypes, microarrays, and fluorescence in situ hybridization (FISH) collected as an amniocentesis and/or chorionic villus sampling (CVS). Genetic screening, such as non-invasive prenatal screening and state analyte screening were not included.

Categories of CHD diagnoses were assigned based on the International Nomenclature for Congenital Heart Surgery (INCHS) by two providers specialized in fetal echocardiography and pediatric cardiology [16]. CHD were classified as one of the following: septal defects, pulmonary venous anomalies, systemic venous anomalies, right heart lesions, left heart lesions, single ventricle, transposition of the great arteries, and thoracic arteries or veins, which includes aortic arch, coronary artery, and ductus arteriosus anomalies.

Each diagnosis was further categorized by severity, with Category 1 indicating low risk of hemodynamic instability in the delivery room, Category 2 minimal risk of hemodynamic instability but requiring postnatal surgical intervention, Category 3 with likely hemodynamic instability requiring immediate specialty care, and Category 4 with expected hemodynamic instability requiring immediate surgical intervention.

Descriptive analysis including frequencies, means, medians, standard deviations, and interquartile range were conducted for all maternal variables, CHD diagnoses, and neonatal variables. Continuous variables were analyzed using the Student t-test and categorical variables were analyzed using the Fisher Exact test.

## Results

Of the 163 twin pregnancies identified, 87 were MC twins (53.3%). Twenty-three MC twin pairs (46 neonates) had CHD (n = 5 concordant CHD, n = 18 discordant CHD pairs) (Fig 1). Table 1 displays the spectrum of CHD in discordant and concordant groups for this population. The most common lesions were septal defects (56% and 45.5% in concordant and discordant cohort) and right heart lesions (40% and 18.2% in concordant and discordant cohort). Lesions present in only the discordant pairs were systemic venous anomalies (4.5%), left heart lesions (9.1%), and thoracic arteries and veins (13.6%). There was no difference in the spectrum or severity of CHD between the two groups (p>0.05).

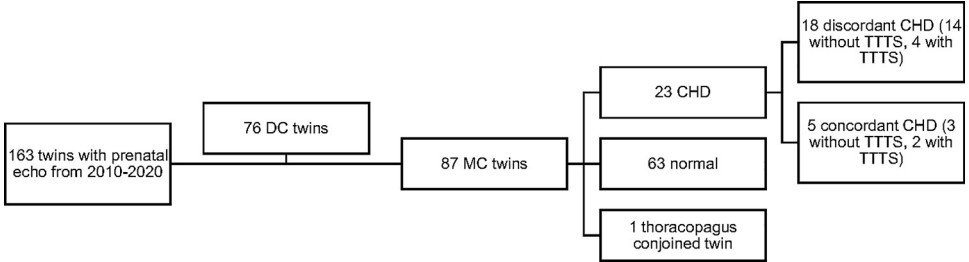

**Fig 1. Study population of twins evaluated.** DC, dichorionic; MC, monochorionic; CHD, congenital heart defects; TTTS, twin-to-twin transfusion syndrome.

**Table 1. Spectrum of CHD in concordant and discordant MC twins, as diagnosed by prenatal echocardiography.**

| Diagnosis[1,2] | Concordant CHD Cohort (n = 10)[3] | Discordant CHD Cohort (n = 22)[3] |
|---|---|---|
| Septal defects | 6 (60%) | 10 (45.5%) |
| Systemic venous anomalies | 0 (0%) | 1 (4.5%) |
| Right heart lesions | 4 (40%) | 4 (18.2%) |
| Left heart lesions | 0 (0%) | 2 (9.1%) |
| Transposition of the great arteries | 0 (0%) | 2 (9.1%) |
| Thoracic arteries/veins | 0 (0%) | 3 (13.6%) |
| CHD severity category[4] | 1 (0) | 1 (0)[5] |

Data are no. (%) or median (IQR).

[1]Based on the International Nomenclature for Congenital Heart Surgery.

[2]p = 0.53.

[3]Includes only neonates affected by CHD.

[4]Category 1 = low risk of hemodynamic instability in the delivery room, Category 2 = minimal risk of hemodynamic stability but requiring postnatal surgical intervention, Category 3 = likely hemodynamic instability requiring immediate specialty care, Category 4 = expected hemodynamic instability requiring immediate surgical intervention.

[5]p = 1.

There was no difference in maternal demographic and risk factors in the two groups based on maternal co-morbidities or known risk factors for CHD, including pre-pregnancy BMI, diabetes, hypertensive disorders of pregnancy, and others (p>0.05) (Table 2). Antepartum length of stay (LOS) on labor and delivery was notably longer for the mothers in the discordant group than those in the concordant group, 10.61 versus 5.20 days, but this difference was not significant (p = 0.101). Five (21.7%) in-vitro fertilization (IVF) pregnancies were identified in this population, all of which were diagnosed with discordant heart lesions. Family history of CHD was rare in both cohorts, with none identified in the concordant and 11.1% in the discordant twin pair group.

Diagnostic genetic testing results were available for 80% of the concordant and 50% of the discordant cohort. There was one abnormal result in each group (20% concordant, 5.6% discordant), with no difference in rate of abnormal genetic results between the two groups (p = 0.395).

Details of the CHD diagnoses, neonatal clinical, genetic, and outcome data with associated extra-cardiac malformations for this population are listed in Tables 3–5. CHD severity was similar in both cohorts, with a median CHD severity category of 1 for both concordant and discordant twins. In the concordant cohort, one twin pair with right heart lesions had a *NOTCH1* variant c.5720C>T and thumb abnormalities identified for both neonates. In the discordant cohort, renal anomalies were identified in three neonates, one twin pair with a VSD/normal phenotype, and one fetus with a PDA and PFO. One neonate with complete atrioventricular canal and pulmonary atresia had intestinal malrotation and heterotaxy syndrome with asplenia, consistent with right atrial isomerism. Overall, rates of malformations in our discordant cohort were low. None of the patients with malformations and CHD had abnormal family history or genetic testing. Placental malperfusion and/or vascular malformations were common in our study, with 4/5 (80%) of concordant and 9/18 (50%) of discordant pregnancies with either cord or placental findings.

Six (25%) twin pairs in the population were complicated by TTTS: 2 concordant (septal defects) and 4 discordant (thoracic arteries and veins 25%, septal defect 25%, systemic venous anomaly 13%, and right heart lesion 13%) (Table 5). All recipient twins were diagnosed with a

**Table 2. Maternal demographics in concordant and discordant MC twins (p = 0.634).**

| Variable | Concordant CHD cohort (N = 5) | Discordant CHD cohort (N = 18) | P-value |
|---|---|---|---|
| Maternal age (years) | 30.40 ± 3.71 | 32.39 ± 7.51 | 0.426 |
| Race | | | 0.745 |
| American Indian or Alaska Native | 0 (0%) | 0 (0%) | |
| Asian | 1 (20%) | 1 (5.6%) | |
| Black or African American | 0 (0%) | 0 (0%) | |
| Native Hawaiian or Other Pacific Islander | 0 (0%) | 0 (0%) | |
| White | 4 (80%) | 14 (77.8%) | |
| Other | 0 (0%) | 2 (11.1%) | |
| Denied | 0 (0%) | 1 (5.6%) | |
| Ethnicity | | | 1.00 |
| Hispanic or Latino | 1(20%) | 4 (22.2%) | |
| Not Hispanic or Latino | 4 (80%) | 13 (72.2%) | |
| Other | 0 (0%) | 1 (5.6%) | |
| Pre-gravid BMI (kg/m$^2$) | 25.70 ± 5.62 | 24.75 ± 6.26 | 0.754 |
| Pre-gestational Diabetes Mellitus | 0 (0%) | 0 (0%) | |
| Gestational Diabetes | 0 (0%) | 4 (22.2%) | 0.539 |
| Chronic Hypertension | 0 (0%) | 1 (5.6%) | 1.00 |
| Hypertensive Disorders of Pregnancy | 2 (40%) | 4 (22.2%) | 0.576 |
| Urinary tract infection | 1 (20%) | 0 (0%) | 0.217 |
| Family History of CHD | 0 (0%) | 2 (11.1%) | 1.00 |
| IVF pregnancy | 0 (0%) | 5 (27.8%) | 0.545 |
| Maternal length of stay (L&D) | 5.20 ± 1.30 | 10.61 ± 13.05 | 0.101 |
| First trimester exposure to SSRI | 1 (20%) | 1 (5.6%) | 0.395 |
| Illicit drug use in pregnancy | 0 (0%) | 0 (0%) | |
| Diagnostic genetic testing | 4 (80%) | 9 (50%) | 0.339 |
| Abnormal genetic results | 1 (20%) | 1 (5.6%) | 0.395 |

BMI, body mass index; IVF, in-vitro fertilization; SSRI, selective serotonin reuptake inhibitor.

Data are mean ± s.d. or no. (%).

heart lesion and 4/6 (67%) donor twins were also affected by CHD. There were no abnormal genetic test results in the twin pairs complicated by TTTS.

Neonatal outcome was analyzed based on concordance of CHD (Table 6). The average gestational age of delivery in both cohorts was ~33 weeks of gestation, with birthweight of 2007g in the concordant and 1960g in the discordant group. The majority of patients, 80% in the concordant cohort and 83% in the discordant cohort, were delivered via cesarean. Almost all neonates were hospitalized in the neonatal intensive care unit (NICU), with a median LOS of 32 (22.75) days in the concordant and 22 (47) days in the discordant cohorts. There was no difference in any neonatal outcomes studied between the two groups (p>0.05) (Table 6). When neonatal outcomes were analyzed to only include neonates affected by CHD in concordant (n = 10) and discordant (n = 22) pairs (Table 7), there was no difference in outcomes, though there was a greater gap between the number of surgeries in the discordant (0.86 ± 1.67) compared to the concordant group (0.40 ± 0.84 surgeries) (Table 7).

## Discussion

We compared a modern cohort of MC twins discordant and concordant for CHD to describe the spectrum of heart lesions identified in these two groups and to detail maternal

**Table 3. Clinical details and outcome data with associated extra-cardiac malformations for monochorionic twins concordant for CHD (excluding those with TTTS).**

| Chorionicity/ amnionicity | Gestation age at birth | Birth weight (g) | Mode of Delivery | APGARs (1 min, 5min) | CHD Category | INCHS detail[16] | INCHS category[16] | Extra-cardiac malformations | Number of surgeries in 1st year | Prenatal Genetics* | Postnatal Genetics | Placental Pathology | Outcome |
|---|---|---|---|---|---|---|---|---|---|---|---|---|---|
| Mono/di | 35w1d | 2620 | Cesarean | 9,9 | 1 | VSD, multiple | Septal defect | None | 0 | Normal karyotype, microarray | None | Foci intervillous thrombohematoma <10% volume | Live birth |
| | | 3325 | Cesarean | 8,9 | 1 | VSD, single | Septal defect | None | 0 | Normal karyotype, microarray | None | Foci intervillous thrombohematoma <10% volume | Live birth |
| Mono/di | 33w3d | 1638 | Vag-Spont | 8,8 | 2 | Pulmonary atresia, VSD | Right heart lesion | Hypoplastic first metacarpal and proximal phalanx of right thumb | 2 | Normal karyotype, microarray, FISH 22q11.2 | Exome sequencing heterozygous NOTCH1 c.5720C>T variant, maternal allele | None | Live birth |
| Mono/di | 33w3 | 1742 | Vag-Spont | 8,9 | 2 | TOF | Right heart lesion | Bifid thumb | 2 | Normal karyotype, microarray, FISH 22q11.2 | Exome sequencing heterozygous NOTCH1 c.5720C>T variant, maternal allele | None | Live birth |
| Mono/di | 32w6d | 1700 | Cesarean | 8,9 | 1 | Tricuspid disease, non Ebstein's; Mitral regurgitation | Right heart lesion | Right inguinal hernia | 0 | None | None | None | Live birth |
| Mono/di | 32w6d | 1850 | Cesarean | 8,9 | 1 | Tricuspid disease, non Ebstein's | Right heart lesion | None | 0 | None | None | Hypercoiled cord, mild chronic deciduitis | Live birth |

*prenatal genetics = diagnostic testing (amniocentesis or chorionic villus sampling).

**Table 4. Clinical details and outcome data with associated extra-cardiac malformations for monochorionic twins discordant for CHD (excluding those with TTTS).**

| Chorionicity/ amnionicity | Gestation age at birth | Birth weight (g) | Mode of Delivery | APGARs (1 min/ 5min) | CHD Category | INCHS detail[16] | INCHS category[16] | Extra-cardiac malformations | Number of surgeries in 1st year | Prenatal Genetics* | Postnatal Genetics | Placental Pathology | Outcome |
|---|---|---|---|---|---|---|---|---|---|---|---|---|---|
| Mono/mono | 28w3d | 820 | Cesarean | 1,6 | 1 | VSD, multiple | Septal defect | Duplicated left renal collecting system, mild hydronephrosis | 0 | Normal karyotype, FISH (13, 18, 21) | Normal karyotype, microarray | None | Live birth |
| | | 1160 | Cesarean | 8,8 | N/A | N/A | N/A | Mild L hydronephrosis | 0 | Normal karyotype, FISH (13, 18, 21) | Normal karyotype, microarray | None | Live birth |
| Mono/di | 31w0d | 1385 | Vag-Spont | 6,8 | 2 | Aortic stenosis, valvar; Mitral stenosis, supravalvar mitral ring | Left heart lesion | None | 2 | Normal karyotype, microarray | Normal karyotype, microarray | 420 g, <10%le weight for GA | Live birth |
| | | 1740 | Vag-Spont | 6,8 | N/A | N/A | N/A | None | 0 | Normal karyotype, microarray | None | 420 g, <10%le weight for GA | Live birth |
| Mono/di | 32w2d | 1595 | Cesarean | 8,9 | N/A | N/A | N/A | None | 0 | Normal karyotype | Microarray 1.5Mb copy loss involving 15q13.2q-13.3 and two copy gains on 7q31.31 and Xp22.2 | Hypocoiled cord | Live birth |
| | | 850 | Cesarean | 7,8 | 1 | ASD, secundum | Septal defect | Left inguinal hernia | 0 | Normal karyotype | Microarray 1.5Mb copy loss involving 15q13.2q-13.3 and two copy gains on 7q31.31 and Xp22.2 | Hypocoiled cord | Live birth |
| Mono/di | 32w6d | 2230 | Cesarean | 9,9 | 1 | ASD, secundum; pulmonary artery stenosis, branch central | Septal defect | None | 0 | None | None | None | Live birth |
| | | 2120 | Cesarean | 9,9 | N/A | N/A | N/A | None | 0 | None | None | None | Live birth |

*(Continued)*

Table 4. (Continued)

| Chorionicity/ amnionicity | Gestation age at birth | Birth weight (g) | Mode of Delivery | APGARs (1 min/ 5min) | CHD Category | INCHS detail[16] | INCHS category[16] | Extra-cardiac malformations | Number of surgeries in 1st year | Prenatal Genetics* | Postnatal Genetics | Placental Pathology | Outcome |
|---|---|---|---|---|---|---|---|---|---|---|---|---|---|
| Mono/di | 34w0d | 1330 | Cesarean | 9,10 | N/A | N/A | N/A | None | 0 | Microarray uniparental disomy 2 but PCR of C2 normal biparental inheritance | None | 75% vascular distribution | Live birth |
| | | 2065 | Cesarean | 8,9 | 1 | Tricuspid valve disease, non Ebstein's related | Right heart lesion | None | 0 | Microarray uniparental disomy 2 but PCR of C2 normal biparental inheritance | None | 25% vascular distribution, hypercoiled, velamentous insertion, infarction and intervillous thrombohematoma | Live birth |
| Mono/di | 34w1d | 1970 | Cesarean | 9,9 | N/A | N/A | N/A | None | 0 | None | None | Bivascular cord | Live birth |
| | | 1760 | Cesarean | 6,8 | 1 | VSD, single | Septal defect | None | 0 | None | None | Normal | Live birth |
| Mono/di | 34w6d | 2300 | Cesarean | 8,7 | 1 | VSD, single | Septal defect | None | 0 | None | None | None | Live birth |
| | | 2315 | Cesarean | 8,9 | N/A | N/A | N/A | None | 0 | None | None | None | Live birth |
| Mono/di | 35w4d | 2380 | Cesarean | 9,9 | N/A | N/A | N/A | None | 0 | None | None | None | Live birth |
| | | 2350 | Cesarean | 9,9 | 1 | Tricuspid valve disease, non Ebstein's related | Right heart lesion | None | 0 | None | None | Hypocoiled cord | Live birth |
| Mono/di | 36w4d | 2375 | Cesarean | 8,8 | 3 | DORV, VSD type | Transposition of the great arteries | None | 5 | Unknown | Normal karyotype, microarray, exome sequencing | Size >90%ile | Death at 3 months of age |
| | | 2680 | Cesarean | 9,9 | N/A | N/A | N/A | None | 0 | Unknown | None | Size >90%ile | Live birth |
| Mono/di | 36w5d | 2220 | Cesarean | 8,8 | 2 | AVC, complete; pulmonary atresia | Septal defect | Intestinal malrotation, hydronephrosis, heterotaxy syndrome with asplenia | 6 | Normal karyotype, microarray | Normal karyotype, microarray, exome sequencing | None | Live birth |
| | | 2260 | Cesarean | 8,9 | N/A | N/A | N/A | None | 0 | Normal karyotype, microarray | None | None | Live birth |

(Continued)

Table 4. (Continued)

| Chorionicity/ amnionicity | Gestation age at birth | Birth weight (g) | Mode of Delivery | APGARs (1 min/ 5min) | CHD Category | INCHS detail[16] | INCHS category[16] | Extra-cardiac malformations | Number of surgeries in 1st year | Prenatal Genetics* | Postnatal Genetics | Placental Pathology | Outcome |
|---|---|---|---|---|---|---|---|---|---|---|---|---|---|
| Mono/di | 37w4d | 2610 | Vag-Spont | 1,1 | 1 | Systemic venous anomalies | Thoracic arteries and veins | None | 0 | None | None | Single UA, marginal insertion, 30% chorionic plate | Live birth |
| | | 3660 | Vag-Spont | 8,9 | 1 | Tricuspid valve disease, non Ebstein's related | Right heart lesion | None | 0 | None | None | 70% chorionic plate | Live birth |
| Mono/di | 38w2d | 2380 | Cesarean | 9,9 | N/A | N/A | N/A | None | 0 | None | None | None | Live birth |
| | | 2995 | Cesarean | 9,9 | 1 | VSD, multiple | Septal defect | None | 0 | None | None | None | Live birth |
| Mono/mono | 34w0d | 1990 | Cesarean | 8,9 | 1 | AVC, complete | Septal defect | None | 0 | None | Normal karyotype, microarray | Single UA | Live birth |
| | | 1930 | Cesarean | 8,9 | 2 | DORV, VSD type | Transposition of the great arteries | None | 2 | None | Normal karyotype, microarray | None | Live birth |
| Mono/di | 35w4d | 2167 | Cesarean | 9,9 | 2 | HLHS | Left heart lesion | None | 2 | None | None | None | Live birth |
| | | 2000 | Cesarean | 9,9 | N/A | N/A | N/A | None | 0 | None | None | None | Live birth |

*prenatal genetics = diagnostic testing (amniocentesis or chorionic villus sampling).

**Table 5. Clinical details and outcome data with associated extra-cardiac malformations for monochorionic twins with TTTS.**

| Chorionicity/ amnionicity | TTTS Stage | Gestation age at birth | Birth weight (g) | Donor or Recipient | Treatment | EGA at treatment | Mode of Delivery | APGARs (1 min/ 5min) | CHD type | INCHS detail[16] | INCHS category[16] | Extra-cardiac malformations | Number of surgeries in 1st year | Prenatal Genetics* | Postnatal Genetics | Placental Pathology | Outcome |
|---|---|---|---|---|---|---|---|---|---|---|---|---|---|---|---|---|---|
| Mono/di | Stage 2 | 34w0d | 1983 | Recipient | Cerclage and laser ablation | 133 | Cesarean | 9,9 | 1 | ASD, secundum | Septal defect | None | 0 | Normal karyotype, microarray | None | Lymphoplasmacytic chronic deciduitis—Intervillous fibrin deposition with villous infarction | Live birth |
| | | | 2105 | Donor | | | Cesarean | 5,7 | 1 | ASD, secundum | Septal defect | None | 0 | Normal karyotype, microarray | None | | Live birth |
| Mono/di | Stage 1 | 33w4d | 1721 | Donor | Amnio-reduction | 174 | Cesarean | 7,8 | 1 | ASD, sinus venosus | Septal defects | None | 0 | Normal karyotype | None | Velamentous insertion, hypercoiled cord, multifocal accelerated villous maturation, chorangioma 0.8 cm, >97%ile size | Live birth |
| | | | 1393 | Recipient | | | Cesarean | 9,9 | 1 | ASD, secundum | Septal defects | None | 0 | Normal karyotype | None | >97%ile size | Live birth |
| Mono/di | Stage 1 | 24w4d | 560 | Recipient | None | N/A | Vag-Spont | 5,7 | 1 | Patent ductus arteriosus | Thoracic arteries and veins | Parietal porencephalic cysts | 1 | None | None | Vasa previa | Death at 7 weeks of age |
| | | | 570 | Donor | | | Vag-Spont | 6,8 | 1 | Secundum, ASD | Septal defects | None | 1 | None | None | None | Live birth |
| Mono/di | Stage 2 | 34w1d | 1785 | Recipient | 2 Laser ablations | 128, 135 | Cesarean | 9,9 | 1 | Persistent left superior vena cava (PLSV) | Systemic venous anomaly | None | 0 | Normal karyotype, microarray | None | Unknown | Live birth |
| | | | 2035 | Donor | | | Cesarean | 8,9 | 1 | Tricuspid valve disease, non Ebstein's | Right heart lesion | None | 0 | Normal karyotype, microarray | None | Unknown | Live birth |
| Mono/di | Stage 4 | 35w4d | 2215 | Donor | None | N/A | Cesarean | 8,8 | N/A | N/A | N/A | None | 0 | None | None | None | Live birth |
| | | | 3010 | Recipient | | | Cesarean | 7,7 | 1 | ASD, secundum | Septal defect | None | 0 | None | None | None | Live birth |
| Mono/di | Stage 1 | 31w0d | 1285 | Donor | Laser ablation | 173 | Cesarean | 1,7 | N/A | N/A | N/A | None | 0 | Normal karyotype, microarray | None | None | Live birth |
| | | | 1460 | Recipient | | | Cesarean | 1,7 | 1 | Patent ductus arteriosus, PFO | Thoracic arteries and veins | Bilateral hydronephrosis, mild right renal stenosis | 0 | Normal karyotype, microarray | None | None | Live birth |

*prenatal genetics = diagnostic testing (amniocentesis or chorionic villus sampling).

**Table 6. Neonatal outcomes and clinical details in concordant and discordant CHD groups.**

| Variable | Concordant CHD cohort (N = 10) | Discordant CHD cohort (N = 36) | P-value |
|---|---|---|---|
| Sex | | | 0.725 |
| Female | 4 (40.00%) | 18 (50%) | |
| Male | 6 (60.00%) | 18 (50%) | |
| Gestational age at birth | 33w5d ± 5d | 33w5d ± 3d | 0.919 |
| Birthweight (g) | 2007.70 ± 568.34 | 1959.92 ± 677.32 | 0.825 |
| Mode of Delivery | | | 1.00 |
| Vaginal | 2 (20.00%) | 6 (16.7%) | |
| Cesarean delivery | 8 (80.00%) | 30 (83.3%) | |
| APGAR 1 minute | 7.90 ± 1.20 | 7.20 ± 2.46 | 0.215 |
| APGAR 5 minute | 8.60 ± 0.70 | 8.20 ± 1.51 | 0.234 |
| NICU Admission | 10 (100.0%) | 31 (86.1%) | 0.570 |
| NICU LOS | 32 (22.75) | 22 (47) | 0.827 |
| Number of surgeries in first year | 0.40 ± 0.84 | 0.53 ± 1.36 | 0.718 |
| Outcome | | | 1.00 |
| Live Birth | 10 (100%) | 34 (94.4%) | |
| Neonatal Death (<28 days) | 0 (0%) | 0 (0%) | |
| Infant Death (>28 days) | 0 (0%) | 2 (5.6%) | |

NICU = neonatal intensive care unit, LOS = length of stay, Data are mean ± s.d., no. (%), or median (IQR).

demographic and pregnancy outcomes. The spectrum of CHD described in this study includes septal defects, systemic venous anomalies, right heart lesions, left heart lesions, transposition of the great arteries (TGA), and thoracic arteries/veins, with the most common lesions in both

**Table 7. Neonatal outcomes and clinical details in concordant and discordant CHD groups for affected neonates only.**

| Variable | Concordant CHD cohort (N = 10) | Discordant CHD cohort (N = 22) | P-value |
|---|---|---|---|
| Sex | | | 0.711 |
| Female | 4 (40.00%) | 11 (50%) | |
| Male | 6 (60.00%) | 11 (50%) | |
| Gestational age at birth | 33w5d ± 5d | 33w4d ± 3w5d | 0.740 |
| Birthweight (g) | 2007.70 ± 568.34 | 1960.32 ± 792.91 | 0.850 |
| Mode of Delivery | | | 1.00 |
| Vaginal | 2 (20.00%) | 5 (22.7%) | |
| Cesarean delivery | 8 (80.00%) | 17 (77.3%) | |
| APGAR 1 minute | 7.90 ± 1.20 | 6.77 ± 2.60 | 0.103 |
| APGAR 5 minute | 8.60 ± 0.70 | 7.86 ± 1.78 | 0.104 |
| NICU Admission | 10 (100%) | 19 (86.4%) | 0.534 |
| NICU LOS | 32 (22.75) | 23.5 (49.75) | 0.471 |
| Number of surgeries in first year | 0.40 ± 0.84 | 0.86 ± 1.67 | 0.305 |
| Outcome | | | 1.00 |
| Live Birth | 10 (100%) | 20 (90.9%) | |
| Neonatal Death (<28 days) | 0 (0%) | 0 (0%) | |
| Infant Death (>28 days) | 0 (0%) | 2 (9.1%) | |

NICU = neonatal intensive care unit, LOS = length of stay, Data are mean ± s.d., no. (%), or median (IQR).

concordant and discordant cohorts being septal defects (60% and 45.5% in concordant and discordant cohort) and right heart lesions (40% and 18.2% in concordant and discordant cohort).

There was similar severity in presentation of CHD, as evidenced by equivalent median CHD severity category scores of affected fetuses in the concordant and discordant pregnancies. AlRais et al. described the spectrum of CHD in a discordant monochorionic cohort of 29 pregnancies, with 10% septal defects and 14% right heart lesions, comparable to our findings. In their cohort, all discordant twin pairs had one affected and one unaffected twin [5]. The majority of our non-TTTS discordant cohort (12/14, 86%) had one normal twin, though in the TTTS discordant subgroup, 50% of pregnancies resulted in both twins with CHD of differing type and severity. These results are consistent with existing data demonstrating that septal defects, are more common in MC twins than in singletons and abnormal blood flow due to TTTS preferentially affects cardiovascular function of the right ventricle in MC twins with associated development of right heart anomalies such as pulmonary atresia [17]. Defects known to be more significantly influenced by genetics, such as Tetralogy of Fallot (TOF), are reported to be equally prevalent in MC twins and singletons [1, 2, 18].

Although somatic mosaicism and somatic chromosomal abnormalities have been described in MC twins, we found no evidence for genetic discordance in twin pairs with discordant CHD [19]. This fact, coupled with the higher rates of discordant (78.2%) versus concordant (21.7%) CHD, low incidence of family history of CHD, and low incidence of genetic abnormalities (8.7%) in this population, suggests that genetic differences in coding regions of the genome are non-contributary in many cases of discordant CHD. There was one abnormal test result in our discordant cohort, a twin pair with a septal defect and left inguinal hernia in one twin and normal finding in the other twin. Their microarray had a 1.5 Mb copy loss of 15q13.2q-13.3 and two copy gains on 7q31.31 and Xp22.2. The copy gains are of unknown significance. The copy loss on chromosome 15 has been described in patients with neurodevelopmental defects but not cardiac abnormalities [20]. The yield of genetic testing in detecting CHD-specific mutations in discordant twins, especially those affected by TTTS, is low, as demonstrated by our negative findings and by studies of copy number variant and exosome sequencing analysis in discordant MC twins [4, 21]. On the contrary, genetic tests with high resolution can be diagnostic in concordant pairs, as evidenced by the *NOTCH1* gene variant associated with non-syndromic TOF that was seen in the twin pairs with right heart lesion and thumb malformations [22]. The utility of in-depth genetic analysis has also been demonstrated in singletons, especially in patients with complex and syndromic CHD [23].

We provide hypothesis generating data that discordant CHD pairs may benefit from tests that provide insight into gene regulation. Although our study did not analyze epigenetic influences, it is important to note that certain post-translational modifications may be significant in the mechanism of discordant CHD development in MC twins. For example, epigenetic analyses found differentially methylated regions in promoters for genes involved in cardiac development in discordant twins [3]. Abu-Halima et al. recently found that microRNAs involved in molecular transport and adrenergic signaling in cardiomyocytes are enriched in CHD compared to non-CHD twins [23].

The etiological question remains how monochorionic twins, presumed to have an identical genotype, can display discordant phenotypes for congenital malformations such as CHD. Our data supports the conclusion that environmental influences likely play a significant role in the pathophysiology of discordant CHD. Factors affecting normal growth and development leading to CHD are hypothesized to occur as early as the first weeks of life, supported by high rates of cardiac looping and laterality defects in discordant CHD as well as the higher prevalence of CHD in twins conceived by assisted reproductive technology [5, 15]. In the discordant cohort,

27.8% had undergone the process of IVF compared to 0% of our concordant cohort, highlighting the influence of gamete manipulation, culture, and embryo implantation on non-familial CHD. In later gestation, although MC twins share a placenta, they do not necessarily experience identical growth environments. Anastomoses between fetal circulations play a significant role in intrauterine development and have the potential to cause syndromes such as TTTS [24]. In this population, the high frequency of TTTS (26%), with incidence of discordance twice that of concordance, indicates the critical role of hemodynamics in the pathophysiology of discordant CHD. The growth conditions for the recipient twin are significantly different than that of the donor twin, subsequently influencing cardiac development. A study analyzing ventricular strain changes in MC twins affected by TTTS shows that at all Quintero stages, recipient left ventricular strain was reduced compared with donors [25]. In stages 3 and 4, recipient right ventricular strain was reduced compared with donors [25]. The relative risk of right ventricular outflow tract obstruction is 70 times that of singletons for MC twins with TTTS, emphasizing the likely importance of hemodynamics on the development of cardiac structures in the second trimester, even after the heart has fully formed [1].

The strengths of this study include a large sample size of discordant twins with a novel comparison to a concordant twin cohort to elucidate environmental risk factors for CHD based on multiple maternal factors and neonatal outcomes. Our study includes a modern cohort with diagnostic testing results achieved using current genetic technology. We were able to follow neonates until at least one year of life, given that all were cared for by a single pediatric cardiology center. Although these results enhance our understanding of the possible mechanism for the development of discordant CHD, it is not possible to establish a temporal relationship between clinical influences and disease development. There are several limitations in our study, including use of a cohort from one specialized center, making this sample subject to selection bias. For example, the prevalence of CHD in our cohort of MC twins is 26.4% compared to literature reporting that about 5.9% of MC twins receive a diagnosis of CHD [1]. The analysis of this specialized cohort, however, also strengthens our findings by allowing for complete data collection with excellent follow up. Additionally, this study details genetic analysis for 80% of the concordant and 50% of the discordant cohort but lacks genetic test results for the remainder of the twin pairs, limiting our ability to draw statistically significant comparisons between the two cohorts with regards to contributing genetic abnormalities. We used abnormal prenatal echocardiography as the criteria for enrollment, and given the low sensitivity of this modality especially in obese patients, CHD diagnosed only postnatally would not be included in our study. Although prenatal fetal echocardiography is effective in early identification of major congenital heart defects, some heart defects with the potential to evolve *in utero* may be missed on prenatal echocardiography [26]. Therefore, this study errs on the side of capturing more severe CHD or CHD more easily detected on prenatal ultrasound. Future studies analyzing both prenatal and postnatal echocardiography records of twins who did not receive a prenatal diagnosis may be helpful in obtaining a representative population of MC twins with discordant CHD.

Our findings emphasize the importance of evaluating MC twins for CHD, especially those conceived through assisted reproductive technology or affected by TTTS. Increasing availability of genetic tests that are sensitive to discrepancies in epigenetic programming are needed in discordant CHD, given the low yield of traditional sequencing analyses for detecting cardiac specific alterations. Diagnostic evaluation that can detect early changes in twin cardiac hemodynamics that are known risk factors for discordant CHD may also improve prenatal detection of cardiac malformations that have implications for neonatal morbidity and mortality. We describe genetic evaluation and fetal and maternal clinical variables in a modern cohort of MC twins to demonstrate the significance of environmental influences and hemodynamics on the

development of discordant CHD, elucidating target areas for early detection and risk modification.

## Author Contributions

**Conceptualization:** Yalda Afshar.

**Data curation:** Helia Imany-Shakibai, Ophelia Yin, Matthew R. Russell, Mark Sklansky, Gary Satou, Yalda Afshar.

**Formal analysis:** Helia Imany-Shakibai, Ophelia Yin, Yalda Afshar.

**Investigation:** Ophelia Yin, Matthew R. Russell, Yalda Afshar.

**Methodology:** Ophelia Yin, Yalda Afshar.

**Project administration:** Yalda Afshar.

**Resources:** Yalda Afshar.

**Supervision:** Mark Sklansky, Gary Satou, Yalda Afshar.

**Validation:** Yalda Afshar.

**Visualization:** Yalda Afshar.

**Writing – original draft:** Helia Imany-Shakibai, Yalda Afshar.

**Writing – review & editing:** Helia Imany-Shakibai, Ophelia Yin, Matthew R. Russell, Mark Sklansky, Gary Satou, Yalda Afshar.

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
