## [Decision Letter · Decision Letter 0]

23 Feb 2021

PONE-D-21-01048

Discordant Congenital Heart Defects in Monochorionic Twins: Risk Factors and Proposed Pathophysiology

PLOS ONE

Dear Dr. Afshar,

Thank you for submitting your manuscript to PLOS ONE. After careful consideration, we feel that it has merit but does not fully meet PLOS ONE’s publication criteria as it currently stands. Therefore, we invite you to submit a revised version of the manuscript that addresses the points raised during the review process.

We look forward to receiving your revised manuscript.

Kind regards,

Alireza Abdollah Shamshirsaz

Academic Editor

PLOS ONE

Journal Requirements:

3. Please include your tables as part of your main manuscript and remove the individual files. Please note that supplementary tables should be uploaded as separate "supporting information" files.

Additional Editor Comments:

Dear Dr. Afshar and colleagues, thanks for submitting your great manuscript to PLOS ONE. Looking forward to see the revise version soon.

Reviewers' comments:

Reviewer's Responses to Questions

**Comments to the Author**

1. Is the manuscript technically sound, and do the data support the conclusions?

Reviewer #1: Partly

Reviewer #2: Yes

Reviewer #3: Yes

2. Has the statistical analysis been performed appropriately and rigorously? 

Reviewer #1: Yes

Reviewer #2: Yes

Reviewer #3: Yes

3. Have the authors made all data underlying the findings in their manuscript fully available?

Reviewer #1: Yes

Reviewer #2: Yes

Reviewer #3: No

4. Is the manuscript presented in an intelligible fashion and written in standard English?

Reviewer #1: Yes

Reviewer #2: Yes

Reviewer #3: Yes

5. Review Comments to the Author

Reviewer #1: Dear Editor,

Thank you for asking me to review this manuscript.

I have following comments and questions, hoping that they will improve the manuscript further.

Methods:

- Please clarify what type(s) of genetic testing was performed.

Results:

- Line 135-137: Please check the claim of LOS being significantly longer. Reported p value is 0.075.

- CHD Category is a categorical data. It should be reported as median with inter quartile range or min-max.

- Line 155: Please classify what is "placental abnormality"

- Line 171-173: Neonatal LOS were 41.23+/-47.16 and 26.83+/-24.17. Standard deviation is very large. Has the data been tested for "normalicy"? Again, reporting median might be appropriate if the data is not normally distributed.

-Table 1: CHD Severity is a categorical variable. It should be reported as median

- Please clarify terms/groups "affected" and "all"

- Please clarify group numbers. Are these number of fetuses or number of twins? Flow diagram indicated 6 pregnancies with concordant anomaly and 18 pregnancies with discordant anomalies.

- Table3a: I am not sure if a conjoint twin should be included in this study and if it is included it should be counted as concordant. Twins are thoracopagus. I suspect there is only one heart.

- Please clarify outcomes column. Please consider rewording "death at 3 months". I think authors meant "3 months of age".

Reviewer #2: Afshar et al studied frequency and spectrum of CHD in a cohort of MC twins as well as risk factors and proposed pathophysiology in discordant CHDs in MC twins. This study was a retrospective single institution study from 1/2010-3/2020. The most common lesions were septal defects and right heart lesions with no significant difference between the concordant and discordant pairs. Interestingly, there was also no difference in rate of abnormal genetic results between the groups and there was no association between clinical risk factors and development of discordant CHD. Authors concluded that their data may demonstrate possibility of environmental and epigenetic influences in the development of discordant CHD in MC twins.

I first would like to congratulate the authors for nicely structured and well-written manuscript. I agree with the authors that some of their data may be influenced by selection bias given the study was from one specialized center and those who were diagnosed postnatally could not be included in the study, but I have found their study findings thought provoking and promising for a larger, potentially multi-center, study. Genetic testing results were available in 67% of the concordant and 50% of the discordant cohort. Although having genetic testing data >50% of their cohort is impressive, not having genetic testing for the entire cohort, especially for the discordant pairs should also be included as a limitation of the study. I have no other comments for the authors.

Reviewer #3: This is an observational study describing the prevalence of congenital heart defects among monochorionic (MC) twin pregnancies according to whether the CDHs were concordant or discordant, and according to genetic results, assisted reproductive technology and the presence or absence of twin-to-twin transfusion syndrome. The authors reported that. genetic testing was abnormal in 17% of the concordant and 6% of the discordant pairs, with no difference in the rates of abnormal genetic results between the groups. The authors did not find a significant association between clinical risk factors and development of discordant CHD, and should be congratulated in their efforts

Additional comments:

1. The rates of abnormal genetic testing was almost three times higher in the concordant CHD group vs. the discordant one. The difference was not significant most likely because of the sample size (type II error). This finding is intuitive and supports the notion that in concordant MC twins the role of genetic abnormalities in CHDs is important

2. The prevalence of CHD in the cohort of MC pregnancies is almost 28% (24/87). This is not representative of the general population and suggests an important selection bias in the study, which needs to be reemphasized in the limitation section of the study

3. Lines 238 to 240, the mechanisms by which TTTS could increase the prevalence of CHD is unlikely to be related with abnormalities in ventricular strain in the recipient twin, otherwise the prevalence of left-sided lesions would be higher that right-sided lesions. However, it usually is the opposite in the recipient twin.

6. PLOS authors have the option to publish the peer review history of their article (what does this mean?). If published, this will include your full peer review and any attached files.

Reviewer #1: No

Reviewer #2: No

Reviewer #3: No

---

## [Author Response · Author response to Decision Letter 0]

15 Mar 2021

Also included in the cover letter. 

March 6, 2021

Dear PLOS Editorial Team Editorial team,

Please find enclosed our REVISED manuscript entitled Discordant Congenital Heart Defects in Monochorionic Twins: Risk Factors and Proposed Pathophysiology.

As you know, in this manuscript we describe discordant congenital heart disease (CHD) in genetically identical monochorionic twin pairs and propose implications for etiological causes in a large referral center over 10 years. We are so appreciative of the thoughtful comments from all reviewers which has improved this submission significantly. All comments have been addressed in the revised version we are submitting. We believe this study provides hypothesis generating data on the possibility of environmental and epigenetic influences versus genotypic factors in the development of discordant CHD in monochorionic twins. 

All authors had access to relevant aggregated study data and other information and all authors take responsibility for the way in which research findings are presented and published, were fully involved at all stages of publication and presentation development. The author list accurately reflects all substantial intellectual contributions to the research, data analyses, and publication or presentation development. No co-author has any disclosures or any relationships or competing interests relating to the research and its publication or presentation.

As part of PLOS One data availability, aggregate level data is and will be available and, in the event, that individual patient level will be needed, we expect a formal material transfer agreement (MTA) to be submitted with the UCLA IRB as part of the human research board, https://ohrpp.research.ucla.edu/. The UCLA IRB and MTA can be contacted at mirb@research.ucla.edu or (310) 825-5344. 

We have read the instruction for authors and below is a detailed point by point response to each comment, with our responses below. 

We affirm that this manuscript is an honest, accurate, and transparent account of these patients’ clinical course, and that no important aspects of the clinical course have been omitted. We look forward to a favorable response of our work in PLOS One.

Yours sincerely,

Yalda Afshar, MD, PhD

Division of Maternal-Fetal Medicine

Department of Obstetrics and Gynecology

University of California, Los Angeles

200 Medical Plaza, Suite 430; Los Angeles, CA 90095

Phone number: (310) 794-8492; YAfshar@mednet.ucla.edu

Below is a detailed point by point response to each comment, with our responses follow and are in red 

Journal Requirements:

We have confirmed that our manuscript matches the PLOS One style requirements and utilized the templates.

As part of PLOS One data availability, aggregate level data is and will be available and, in the event, that individual patient level will be needed, we expect a formal material transfer agreement (MTA) to be submitted with the UCLA IRB as part of the human research board, https://ohrpp.research.ucla.edu/. The UCLA IRB and MTA can be contacted at mirb@research.ucla.edu or (310) 825-5344. 

 3. Please include your tables as part of your main manuscript and remove the individual files. Please note that supplementary tables should be uploaded as separate "supporting information" files.

Thank you. We have done this in the revision.

Additional Editor Comments:

Dear Dr. Afshar and colleagues, thanks for submitting your great manuscript to PLOS ONE. Looking forward to see the revise version soon.

Reviewers' comments:

Reviewer's Responses to Questions

Comments to the Author

1. Is the manuscript technically sound, and do the data support the conclusions?

Reviewer #1: Partly

Reviewer #2: Yes

Reviewer #3: Yes

2. Has the statistical analysis been performed appropriately and rigorously?

Reviewer #1: Yes

Reviewer #2: Yes

Reviewer #3: Yes

3. Have the authors made all data underlying the findings in their manuscript fully available?

Reviewer #1: Yes

Reviewer #2: Yes

Reviewer #3: No

4. Is the manuscript presented in an intelligible fashion and written in standard English?

Reviewer #1: Yes

Reviewer #2: Yes

Reviewer #3: Yes

5. Review Comments to the Author

Reviewer #1: Dear Editor,

Thank you for asking me to review this manuscript.

I have following comments and questions, hoping that they will improve the manuscript further.

Methods:

- Please clarify what type(s) of genetic testing was performed.

Thank you for this clarifying point. We have updated this in our methodology to say: “Specifically, genetic test modalities included in this analysis were karyotypes, microarrays, and fluorescence in situ hybridization (FISH)” (Line 162-164)

Results:

- Line 135-137: Please check the claim of LOS being significantly longer. Reported p value is 0.075.

Thank you for this comment. We have adjusted this claim to say: “Antepartum length of stay (LOS) on labor and delivery was notably longer for the mothers in the discordant group…” (Line 200-203)

- CHD Category is a categorical data. It should be reported as median with inter quartile range or min-max.

Thank you for bringing this to our attention. We have adjusted Table 1 and the body of the text to report CHD Severity as a median with an interquartile range. 

- Line 155: Please classify what is "placental abnormality"

We included all placental vascular malformations and/or mal-perfusions noted on the pathology report to be a placental abnormality.

- Line 171-173: Neonatal LOS were 41.23+/-47.16 and 26.83+/-24.17. Standard deviation is very large. Has the data been tested for "normalicy"? Again, reporting median might be appropriate if the data is not normally distributed.

Thank you for this comment. We have adjusted Table 4a and 4b to report Neonatal LOS as a median with an interquartile range. These values have also been reported in the text “Almost all neonates were hospitalized in the neonatal intensive care unit (NICU), with a median LOS of 32 (22.75) days in the concordant and 22 (47) days in the discordant cohorts” (Line 247-249)

-Table 1: CHD Severity is a categorical variable. It should be reported as median

Thank you for bringing this to our attention. We have adjusted Table 1 and the body of the text to report CHD Severity as a median with an interquartile range. 

- Please clarify terms/groups "affected" and "all"

We have removed the term “affected” from the last row of this table and instead stated in the footnotes that this table reflects data from only affected neonates. We have also removed the column labeled “CHD severity category (all)” as we felt this data point did not add to our discussion. 

- Please clarify group numbers. Are these number of fetuses or number of twins? Flow diagram indicated 6 pregnancies with concordant anomaly and 18 pregnancies with discordant anomalies.

Thank you for this comment. We have updated our footnotes in Table 1 to clarify that the sample described is neonates affected by CHD. 

- Table3a: I am not sure if a conjoint twin should be included in this study and if it is included it should be counted as concordant. Twins are thoracopagus. I suspect there is only one heart.

Thank you for this clarifying point. Yes, as a thoracopagus there was a single heart with CHD, and we agree with the reviewer that this patient should be excluded. We have excluded this patient, updated our statistics and tables, and included a sentence in the methods section to clarify this exclusion: “Dichorionic and conjoined twins were excluded from this study” (Line 148)

- Please clarify outcomes column. Please consider rewording "death at 3 months". I think authors meant "3 months of age".

We have changed the wording of the outcome for Table 3b row 17 to “death at 3 months of age” and Table 3c row 5 to “death at 7 weeks of age”. 

Reviewer #2: Afshar et al studied frequency and spectrum of CHD in a cohort of MC twins as well as risk factors and proposed pathophysiology in discordant CHDs in MC twins. This study was a retrospective single institution study from 1/2010-3/2020. The most common lesions were septal defects and right heart lesions with no significant difference between the concordant and discordant pairs. Interestingly, there was also no difference in rate of abnormal genetic results between the groups and there was no association between clinical risk factors and development of discordant CHD. Authors concluded that their data may demonstrate possibility of environmental and epigenetic influences in the development of discordant CHD in MC twins.

I first would like to congratulate the authors for nicely structured and well-written manuscript. I agree with the authors that some of their data may be influenced by selection bias given the study was from one specialized center and those who were diagnosed postnatally could not be included in the study, but I have found their study findings thought provoking and promising for a larger, potentially multi-center, study. Genetic testing results were available in 67% of the concordant and 50% of the discordant cohort. Although having genetic testing data >50% of their cohort is impressive, not having genetic testing for the entire cohort, especially for the discordant pairs should also be included as a limitation of the study. I have no other comments for the authors.

Thank you for the comment about the work and for the summary. We agree that a larger multi-center study would be paramount to draw stronger conclusion and we are working on this work in collaboration with multiple centers now that we have obtained results from our own center. So, thank you for this comment. Agree one of the biggest limitations is the lack of universal testing. We have included the lack of available genetic test results for the entire cohort in the limitations section of our discussion (Line 384-387)

Reviewer #3: This is an observational study describing the prevalence of congenital heart defects among monochorionic (MC) twin pregnancies according to whether the CDHs were concordant or discordant, and according to genetic results, assisted reproductive technology and the presence or absence of twin-to-twin transfusion syndrome. The authors reported that. genetic testing was abnormal in 17% of the concordant and 6% of the discordant pairs, with no difference in the rates of abnormal genetic results between the groups. The authors did not find a significant association between clinical risk factors and development of discordant CHD, and should be congratulated in their efforts

Additional comments:

1. The rates of abnormal genetic testing was almost three times higher in the concordant CHD group vs. the discordant one. The difference was not significant most likely because of the sample size (type II error). This finding is intuitive and supports the notion that in concordant MC twins the role of genetic abnormalities in CHDs is important

Thank you for this comment and we agree that we would have loved to have universal genetic testing in the entire cohort to draw a stronger conclusion. In light of your comment, re: role of type 2 error b/c of limitation in the difference in testing availability we have added language to the limitations about the lack of genetic testing in the entire population (Line 384-387). Thank you. 

2. The prevalence of CHD in the cohort of MC pregnancies is almost 28% (24/87). This is not representative of the general population and suggests an important selection bias in the study, which needs to be reemphasized in the limitation section of the study

Thank you so much for this observation. We have emphasized the higher rates of CHD in our cohort of MC twins in the limitation section of the study and compared this prevalence to literature (Line 381-383). 

3. Lines 238 to 240, the mechanisms by which TTTS could increase the prevalence of CHD is unlikely to be related with abnormalities in ventricular strain in the recipient twin, otherwise the prevalence of left-sided lesions would be higher that right-sided lesions. However, it usually is the opposite in the recipient twin.

We have reviewed literature in support of the reviewers comment and our comment (#25) and because of this discrepancy we have decided to remove this theoretical framework from our discussion and we believe it provides clarity to the discussion without this controversial proposed pathophysiology. Thank you.

---

## [Editor Report · Decision Letter 1]

21 Apr 2021

Discordant Congenital Heart Defects in Monochorionic Twins: Risk Factors and Proposed Pathophysiology

PONE-D-21-01048R1

Dear Dr. Afshar,

We’re pleased to inform you that your manuscript has been judged scientifically suitable for publication and will be formally accepted for publication once it meets all outstanding technical requirements.

Kind regards,

Alireza Abdollah Shamshirsaz

Academic Editor

PLOS ONE

Additional Editor Comments (optional):

Well done and please send us more good quality manuscript.
---

## [Editor Report · Acceptance letter]

27 Apr 2021

PONE-D-21-01048R1 

Discordant congenital heart defects in monochorionic twins: risk factors and proposed pathophysiology 

Dear Dr. Afshar:

I'm pleased to inform you that your manuscript has been deemed suitable for publication in PLOS ONE. Congratulations! Your manuscript is now with our production department. 

Kind regards, 

on behalf of

Dr. Alireza Abdollah Shamshirsaz 

Academic Editor

PLOS ONE